# Mechanocardiography-Based Measurement System Indicating Changes in Heart Failure Patients during Hospital Admission and Discharge

**DOI:** 10.3390/s22249781

**Published:** 2022-12-13

**Authors:** Tero Koivisto, Olli Lahdenoja, Tero Hurnanen, Juho Koskinen, Kamal Jafarian, Tuija Vasankari, Samuli Jaakkola, Tuomas O. Kiviniemi, K. E. Juhani Airaksinen

**Affiliations:** 1Department of Computing, University of Turku, 20500 Turku, Finland; 2CardioSignal Inc., 20100 Turku, Finland; 3Heart Center, Turku University Hospital, University of Turku, 20520 Turku, Finland

**Keywords:** accelerometer, gyrocardiography, gyroscope, mechanocardiography, heart failure, seismocardiography, ECG, serial measurements, self-screening

## Abstract

Heart failure (HF) is a disease related to impaired performance of the heart and is a significant cause of mortality and treatment costs in the world. During its progression, HF causes worsening (decompensation) periods which generally require hospital care. In order to reduce the suffering of the patients and the treatment cost, avoiding unnecessary hospital visits is essential, as hospitalization can be prevented by medication. We have developed a data-collection device that includes a high-quality 3-axis accelerometer and 3-axis gyroscope and a single-lead ECG. This allows gathering ECG synchronized data utilizing seismo- and gyrocardiography (SCG, GCG, jointly mechanocardiography, MCG) and comparing the signals of HF patients in acute decompensation state (hospital admission) and compensated condition (hospital discharge). In the MECHANO-HF study, we gathered data from 20 patients, who each had admission and discharge measurements. In order to avoid overfitting, we used only features developed beforehand and selected features that were not outliers. As a result, we found three important signs indicating the worsening of the disease: an increase in signal RMS (root-mean-square) strength (across SCG and GCG), an increase in the strength of the third heart sound (S3), and a decrease in signal stability around the first heart sound (S1). The best individual feature (S3) alone was able to separate the recordings, giving 85.0% accuracy and 90.9% accuracy regarding all signals and signals with sinus rhythm only, respectively. These observations pave the way to implement solutions for patient self-screening of the HF using serial measurements.

## 1. Introduction

Heart failure (HF) is a common clinical syndrome causing morbidity and mortality. It is present in 1–2% of the total population and 11.8% of those aged 65 years or older in developed countries [1]. There are many underlying cardiovascular conditions leading to HF. The leading causes of HF are coronary artery disease, valvular disease, and hypertension [2,3,4,5]. For patients with systolic HF, i.e., HFrEF, there are several medications and device-mediated therapies that improve the prognosis. Diastolic HF, i.e., HFpEF, is a more complex entity, and currently, the therapies focus on the underlying diseases [6,7]. Treatment modalities depend on the etiology of HF; however, medical therapies using ACE inhibitors/ATR blockers, beta-blockers, SGLT2 inhibitors, sacubitril/valsartan, and diuretics are the cornerstones of HF management [8]. Ultrasound analysis is the gold standard for diagnosing HF, and it may be supplemented by blood markers such as B-type natriuretic peptide (proBNP). The prognosis of HF is poor, and 30–40% of patients diagnosed with HF die within a year [9]. One of the main causes of the cost associated with HF is subsequent hospital periods caused by the worsening of the patient’s condition.

Seismocardiography (SCG) [10,11] measures heart-induced movements using a sensor placed on the chest. Ballistocardiography (BCG) measures the overall body movements induced by the heart to a sensor placed on a bed or chair [11]. Sometimes these terms are used interchangeably. Gyrocardiography [12], which is based on gyroscopes, can be used to complement SCG, which is based on accelerometers. The GCG signal has been shown to be more monomorphic than SCG, which is extremely beneficial in the detection of heartbeats either alone or supplemented with SCG [13]. We denote the combination of SCG and GCG as mechanocardiography (MCG), which includes all six acceleration and rotation axes.

Recently, SCG, GCG, and BCG have been proposed to find indications related to the condition of HF patients [14,15,16,17]. In our previous study [15], we examined the detection of acute decompensated HF with a smartphone in interpersonal settings (different patient IDs in separate groups) using only HF patients who suffer from atrial fibrillation (AF). The origin of data set used was in a clinical study aiming to diagnose AF. Here, we extend our previous work to an intrapersonal setting, where each patient is his/her own reference (as in [16]). We use a dedicated Holter device (single-lead ECG) to supplement MCG. An advantage of this setting, among others, is the capability to extract clinically relevant features in combination with applications in patient self-screening using serial (or continuous-time) measurements. The objective of the MECHANO-HF study is to assess which kind of person-specific changes can be observed between the signals acquired during the acute decompensation state and the normal state. Thus, our work builds the grounding for effective screening of HF reducing hospitalizations and the treatment cost in the future.

## 2. Background

Electrocardiography (ECG) and Holter monitoring at home is the gold standard in investigations related to many cardiac diseases such as AF. In HF patients, ECG can be used as a preliminary screening tool for ruling out purposes [18]. If the ECG is normal, it is likely that the patient does not have HF [18]. On the other hand, for instance, the left bundle branch block (LBBB) is a strong sign of HF in ECG [19]. The actual diagnosis and characterization of HF are still performed using ultrasound, which also allows the measurement of the ejection fraction (EF) related to the pumping efficiency of the heart. EF is denoted in percent; a higher EF generally indicates better pumping efficiency of the heart, whereas an EF below 40% indicates HF with reduced ejection fraction (HFrEF). For HF patients with preserved ejection fraction (HFpEF), the EF may be larger than this, but they have other indications of impaired heart function. Regarding ECG and SCG, a recent study indicates that SCG might perform better in terms of correlation to peak oxygen uptake (VO2), which is related to the fitness of the patient [20].

Implantable devices have been designed to alleviate the symptoms and prognosis of HF. Some of these CRT (cardiac resynchronization therapy) devices aim at synchronizing the timing of the heart, whereas others, ICD (implantable defibrillator) devices, can be used to prevent sudden cardiac death or to track the condition of the patient in long-term use [21]. In [21], the correlation between the first heart sound (S1) and EF was observed. Generally, an increased magnitude of S3 (third heart sound) and diminished S1 (first heart sound) are well-known markers of HF [22]. Another aspect is the follow-up of the patient’s weight, as fluid accumulation may happen prior to the patient’s condition’s worsening [23]. However, [24] no advantage from daily follow-up of the patient’s weight in ambulatory settings in reducing hospital admissions and prognosis was found. In addition to the criteria used in [14], changes in acceleration and rotational kinetic energy (and power) have been proposed to characterize HF [16].

Many different embedded device implementations have been developed for wearable sensing and HF management. A device called Shimmer 3 was used in [25] for joint sensing with a 3-axis accelerometer (SCG) and a 3-axis gyroscope (GCG) supplemented with impedance cardiography (ICG). In [26], a real-time wearable system equipped with an accelerometer and ECG was implemented with the target of optimizing power consumption. Regarding devices targeted for HF management in [27], a modified home weighing scale was designed for classification between the HF stages. The effects of posture changes on SCG signals were studied in [28] for HF patients using a wearable patch. Recently, a custom-built, round wearable patch utilizing SCG and ECG was proposed for reducing hospital readmissions due to HF [29].

## 3. Measurement Device

The measurement system of this study was a custom-designed data logger equipped with a 3-axis accelerometer (ADXL355, Analog Devices Inc., Wilmington, MA, USA), 3-axis gyroscope (LSM6DS3, STMicroelectronics, Geneva, Switzerland), and single-lead ECG. The objective was to design a device for making spot measurements. The data logger device was based on a MAXREFDES100 hsensor (Maxim Integrated, San Jose, CA, USA) platform which was modified in order to enable the operation of an ADXL355 together with MAX30003 biopotential AFE (Maxim Integrated, San Jose, CA, USA) and LSM6DS3 inertial measurement sensors. Additionally, a power button on the hsensor platform was replaced with an external button board. The electronics were installed to a custom-made 3D printed Nylon enclosure. The measurement setup where the patient is in a supine position along with the Holter device is shown in Figure 1.

### 3.1. Power Supply and Communication

The device can be operated in a measurement mode only when powered by a battery (CR2032). If the USB is connected, the system will initialize itself to a command mode for data transfer and flash memory erase. The ADXL355 sensor was wired to the I2C peripheral of the MAX32620, and a supply voltage for the ADXL355 was taken from the MAX14720 power management IC’s adjustable buck-boost regulator output. Since the EVAL-ADXL355-PMDZ evaluation board must be supplied with voltage levels between 2.25 and 3.6 V (I/O supply internally wired in the board), and on the other hand, the MAX32620 microcontroller is operated with a 1.8 V supply, there is a certain I/O-level mismatch on the I2C communication. To avoid additional challenging wiring due to bypassing the ADXL355 internal LDOs, removal of the noise suppression ferrite bead, and operation without any isolation between the analog and digital supplies, it was decided to use a supply voltage as low as possible where communication is working reliably but still within the specifications of the MAX14720 buck-boost regulator, ADXL355 sensor, and SML-P11UTT86 LED.

It was observed that communication worked at and below +2.8 V, but not above +3.0 V; therefore, +2.5 V was chosen to maintain a fair trade-off. The downside is that it is below the MAX30101 optical sensor LED driver’s specifications (min. +3.1 V) and below the ratings of the LDO regulator MAX8880, which is dedicated to the MAX30205 human body temperature sensors. The use of another type of ADXL355 evaluation board (EVAL-ADXL355Z) allowing the supply of +1.8 V without additional work had to be rejected because of bad component availability at the time. The average current consumption of the system in the measurement mode was measured with an Agilent 34410A multimeter using a benchtop power supply. The full capacity of nominally 220–240 mAh can be only accessed if discharged at a rate close to the rate stated in battery data sheets (<0.5 mA) before the voltage drops too low (2.0 V is usually referred to as a "dead" battery). Since the functional endpoint (FEP) cannot be lower than +2.0 V in the case of a Holter monitor (preferably +2.1 V) and the discharge current is considerably high, the effective capacity can be somewhere between 80 and 100 mAh, indicating approximately 8.6 to 10.7 h of continuous use.

### 3.2. User Interface of the Device

There are two basic modes in which to operate the Holter monitor: a measurement mode and a command mode. The measurement mode is the nominal mode for the acquisition of sensor data to a flash memory from ADXL355, LSM6DS3, and MAX30003 devices. The measurement mode is initiated and stopped with a press of the power button. The command mode is initiated when a USB is connected and serves as a platform for transferring sensor data to external data storage. Necessary training was given to the nurses and doctors at the hospital to successfully perform the measurements. The main parameters of the device, including some other implementations, are shown in Table 1.

## 4. Clinical Study Information

The MECHANO-HF study—a joint effort of the Heart Center, Turku University Hospital, Turku; and the Department of Computing, University of Turku, Turku, Finland—included 20 patients with both admission and discharge measurements from each. The inclusion criteria for the study were: (1) the patient’s age was at least 18 years, (2) he/she was capable of understanding and willing to sign an informed consent form, and (3) he/she was hospitalized due to a worsening period of HF (either HFrEF or HFpEF).

Data acquisition was performed at the Heart Center, Turku University Hospital. The study protocol was approved by the Medical Ethics Committee at The Hospital District of Southwest Finland. All patients enrolled in the study provided written informed consent for participation. The data analysis part was performed at the Department of Computing, University of Turku, Turku, Finland.

### 4.1. Data Acquisition

The measurements were performed using the described 3-axis SCG, 3-axis GCG, and a single-lead Holter device. The intended use of the device was only for data acquisition using short measurements (i.e., it was not designed for continuous-time use). All recordings were performed in a supine position by placing the sensor on the patient’s chest (to the lower one-third portion of the chest), and the bed angle was kept at approximately 30° (both at admission and at discharge). Each measurement contains 3 accelerometer signals (one for each axis), 3 gyroscope signals (one for each axis), and a single-lead ECG (lead I). The device was attached to the skin using double-sided tape without hair removal. The measured acceleration and angular velocity ranges of the accelerometer and gyroscope were set to +−2 g and +−250 dps, respectively. The accelerometer had a noise density of 25 μg/$\sqrt{H}$z, and the gyroscope rate noise density was 7 mdps/$\sqrt{H}$z. All MCG and ECG data were recorded simultaneously and resampled to have a sampling frequency of 200 Hz. Custom-made software was used to read and process the measurements.

### 4.2. Patient Recordings

Each patient had two recordings: one recorded when arriving at the hospital (i.e., decompensation state measurement, where the patient was in worse condition) and one recorded when the patient was discharged (i.e., the patient was in better condition after the treatment). Diuretic medication (when necessary) was given during the treatment period at the hospital. The time between the recordings of the same person was dependent on the length of the hospital stay; typically, it was a few days. All the measurements were inspected visually before and after applying the automated motion-artifact-removal algorithm.

The mean durations (and standard deviation) of the recordings before and after motion artifact removal were 295.9 (98.9) seconds and 206.5 (78.6) seconds, respectively (i.e., approximately 3.4 min). The minimum and maximum durations of all the recordings were 183 and 512 s before and 85 and 370 s after motion artifact removal. The measurements of patient 8 were of relatively poor quality, but we decided to include those in the analysis due to completeness. The data collection setup enabled pairwise comparison of patients with a reasonable number of measurements.

### 4.3. Demographics Information

The mean weight of the patients on admission was 85.6 (SD = 26.2) kg and on discharge was 84.5 (SD = 28.8, NA = 4) kg. AF (or other significant arrhythmias) was present in the ECG recordings in 8/20 (40%) patients on hospital admission and 9/20 (45%) of patients on hospital discharge. Patients’ EF during admission were 33.9 (SD = 12.3, NA = 4) % using ultrasound. EF at discharge was not measured. In total, there were 12/16 (75.0%) patients whose EF was both measured and below 40 (HFrEF). The mean heart rate (HR) during admission was 86.0 (SD = 19.1) bpm and 82.4 (SD = 17.6) bpm during discharge. In total, HR was decreased in 11/20 (55%) of patients (estimate from ECG signal). The patient information of the study can be found in Table 2.

## 5. Methods

### 5.1. Motion Artifact Removal

Motion artifacts were removed from the signals (SCG, GCG, and ECG) using an automated algorithm. The algorithm was originally developed for a more challenging usage scenario of removing motion artifacts from chest pain (myocardial infarction, STEMI) recordings [30]. It was straightforward to deploy the same approach for this study. The device was the same, and there was no need to tune the noise thresholds again. The specifications for the algorithm are that the result (artifact-free signal) should be always the same for the same input signal, the method needs to be computationally light in order to function in an embedded device, the amount of tunable thresholds should be minimized, and finally, it needs to be able to determine if the overall quality of the measurement is insufficient (e.g., interpreted using the duration of the resulting signal).

The algorithm takes advantage of a separate threshold (same thresholds across all measurements) for each accelerometer and gyroscope axis and ECG. Each signal was divided into 5 s segments (including all axes and ECG) and further swept through with a denser 1 s window. If in this dense window at any axis location exceeded the set threshold, the current 5 s segment was declared to contain an artifact and was removed. The value for the dense window which was compared to the threshold was calculated as signal absolute traversal length applied to the raw signal (i.e., before filtering). The noise thresholds depend on the mean absolute amplitude of each particular axis. Finally, the longest uniform section of segments with no gaps between the segments was selected as the algorithm output (i.e., cleaned output signal). Thus, the length of the cleaned signal was smaller than the length of the original signal. The algorithm input and the algorithm output at all times match the different MCG axes and ECG.

After motion artifact removal, a Butterworth bandpass filter (e.g., 2–90 Hz, lower frequency at least 2 Hz) was applied to remove the breathing and varying baseline.

### 5.2. Feature Extraction

Feature extraction was next performed with features that we previously developed in another study: (1) signal strength, (2) amplitudes and time-intervals (e.g., R-peak to AO and AC), and (3) stability within specific signal windows defined using the R-peaks of ECG (e.g., around different heart sounds, S1–S3) [30]. These features—i.e., their Matlab R2017a implementations—had been developed beforehand, so we did not develop new features based on the data we already had. The features were then organized into groups based on what characteristics they measure (S1 strength, S2 strength, etc.; or stability around these locations).

Beat stability (or correlation) features were used to indicate how a certain part/portion of the cardiac cycle changes over time. A chosen cardiac cycle component was located individually from each cardiac cycle from the signal. These components were then compared to each other using Pearson correlation. When traveling through all beats of the signal at some axis, the correlations between the beats form a correlation matrix. The mean or median value of this matrix was used as the feature. Prior to calculating the correlations, the signal was filtered using a bandpass filter whose range characterizes that particular feature.

Figure 2 shows how the estimation of locations of S1–S3 was performed. S1 (i.e., AO-peak and aortic opening) is located by finding the maximum value inside the S1 window, which starts from the previous R-peak, and its length is one quarter of the current RR interval. Similarly, AC (aortic closure, S2) is located by finding the maximum value inside the S2 window, which starts at one quarter after S1, and its length is one quarter of the current RR interval. S3 strength is the cumulated absolute signal value inside the S3 window, which is located 80 ms after S2, and its length is 125 ms (if the length of the RR interval is 1000 ms). From ECG only, the R-peak locations were used in deriving the features. During the calculation of signal strength features, these windows (related to S1–S3) determine the area where the strength is calculated.

### 5.3. Feature Analysis Method

The data (40 recordings in total) were analyzed in pairs (N = 20); each patient had his/her recording at hospital admission and discharge. After feature extraction, the features were organized as a feature matrix. The feature matrix consisted of 40 rows (one for each measurement) and approximately six hundred columns (one for each feature), and the label data including patient ID and admission/discharge information, were a separate vector. This matrix was further divided into two matrices (by selecting either paired or odd rows) to represent the feature values at admission and the feature values at discharge, respectively. A fourth matrix was then generated by comparing these two matrices, resulting in a binary value of 1 or 0 depending on which matrix had a larger (or smaller) value at the corresponding location.

Thus, pairwise comparisons between the number of increasing/decreasing data pairs for the measurements were conducted. The most promising sets of features (not only a single feature) in terms of their capability to divide the admission and discharge measurement into constantly increasing/decreasing pairs were then found.

In this study, we did not apply supervised machine learning (ML) to the data, as there were significant variations in the metadata parameters (for instance, in terms of the occurrence of AF). On the other hand, when considering sinus rhythm measurements, only the amount of data was not sufficient for the supervised ML approach. Instead, our interest was to find whether any physiological phenomena could be found based on pairwise reasoning supplemented with unsupervised ML. The target was to find multiple similar features showing constant decrease or increase in the pairwise analysis within the data set. Thus, instead of using a supervised classifier, we aimed to find representative groups of features changing in the same direction, which would describe a similar phenomenon.

Assuming there will be more data in the future, the features of this study could be used in classification based on simple logical reasoning (e.g., by taking into account the patient’s rhythm) or via a classifier and cross-validation. We avoided relying only on individual features, since N is small, and they might be therefore sensitive to overfitting.

## 6. Results

In Figure 3, all S3 signal strength and all S1 correlation features are shown. In producing the plot first, each column of the feature matrix was normalized to between zero and one. Then, the median value of all features during admission and the median value of all features during discharge were calculated and plotted on the x-axis and y-axis, respectively. Therefore, the most promising features, the features that change most, are concentrated on the upper left and lower right parts of the plot. Moreover, if admission values are larger than discharge values, the feature will be toward the lower right corner; and if admission values are smaller than discharge values, the features will be located toward the upper left corner. It can be observed that S3 strength features are located dominantly on the lower-right portion of the plot with respect to x=y line. In addition to selected S3 features, the total signal RMS strength (especially of the gyroscope) represents this cluster efficiently.

In the same plot, it can be observed that stability around S1 provides a unique cluster among the features which is located above the x=y line (dotted line). For these features, the admission values tend to be constantly lower than the discharge values. Moreover, the k-means clustering algorithm was also able to distinguish the S1 correlation and S3 strength clusters. In order to represent the correlation cluster efficiently for the purposes of this study, one correlation feature from each axis was selected for further analysis.

Based on pairwise reasoning and the analysis above, three sets of features were selected for further investigation. The first was signal RMS strength, the second was signal third heart sound (S3) strength, and the third was signal stability around the first heart sound (S1). These are discussed in the following in more detail. For the purpose of representing the results compactly, we always selected one representative feature from each of the three categories.

### 6.1. Signal RMS Strength

The first set of features contains the total RMS (root-mean-square) strength of all accelerometers together and all gyroscope axes together. Our prior assumption was that the signal’s strength would be lower at admission since the patient’s EF would be small. However, the situation is the opposite: both in SCG and in GCG, the RMS strength increases in both accelerometer and gyroscope when the patient is in decompensation (see Table 3). With omitting AF cases—in either admission or discharge—it can be observed from Table 3 that accelerometer strength decreased towards discharge in 9/11 (81.8%) cases, and gyroscope decreased towards discharge strength in 10/11 (90.9%) cases. A potential physiological explanation for this is that the heart is forced to increase its efforts in order to compensate for the loss of pumping efficiency (small EF), and this phenomenon can be observed using MCG. Additionally, increased HR during admission can be a contributor to this. An advantage of this feature is that it is ECG-independent (there is no need to align a window, e.g., around any heart sound). The passband for the signal RMS strength presented here was 5–40 Hz, but it is robust to variations in the passband.

### 6.2. Third Heart Sound Strength (S3)

We sought to find the approximate location of S3 using the R-peak of the ECG. The methodology was first to find S1 accurately using R-peak and use it to find S2. Finally, the approximate location of S3 was found using a search window starting 80 ms after the S2 location and lasting 125 ms (up to 205 ms after S2), assuming a heartrate of 60 bpm. Instead of using S3 amplitude, the overall strength of the signal within the window around S3 showed better performance (in the pairwise analysis). The results can be seen in Table 3. It can be observed that the changes in this feature apply not only in the groupwise feature setting but the axis-wise setting also. AF might have some unexpected effects, e.g., to the (automatic) window estimate (around S3), and therefore, the results are reported for an SR-only case also. According to Table 3, in the GyroX and GyroY axes, S3 combined with sinus rhythm reached a similar performance to gyroscope RMS strength in 90.9% of cases. However, the result is slightly improved in comparison with RMS strength if all measurements, including AF, are considered: in 17/20 (85%) of cases, S3 energy is decreased. It is important to note that a similar analysis was performed with features around S1 and S2, and in our case, S3 was shown to outperform these. Thus, an elevated S3 strength seems to be an indicator of the decompensation state during admission in MCG. The differences in the selected signal strength features between admission and discharge can be seen in Figure 4.

### 6.3. Stability of First Heart Sound (S1)

The third discriminating feature was the stability of beat windows around S1. The first R-peak of the ECG was again found, and using it, location S1 was detected. Then, a window starting at 125 ms before S1 and lasting 750 ms (up to 625 ms after S1) was found. Using these (fixed-length) windows, a correlation matrix was constructed based on all extracted beats (for each axis separately). The correlation matrix was constructed based on amplitude-normalized windows. From the correlation matrix, the mean value was calculated as the result. Prior to these, motion artifact removal and subsequent filtering (2–30 Hz passband) were performed in order to reduce the effect of noise. The mean Pearson’s correlation becomes higher (around S1) when the patient’s condition becomes better, i.e., towards discharge (see Table 4). However, this feature with the best-performing axes of AccY and GyroZ axes seems a bit less discriminative than the RMS and S3 strength features. On the other hand, it could be used to supplement the signal strength features; for instance, patient 4 was changed into the opposite direction to the majority using the signal strength, to correct direction with the stability feature.

## 7. Discussion

In comparison with our previous study [15], where an interpersonal setting was used, here we focused on an intrapersonal setting. Instead of a smartphone, in this study, we used a dedicated monitoring device including ECG also, which facilitates clinical interpretation of the features. It is important that the features which are useful in MCG can be interpreted by cardiologists, who will eventually make the decision to recommend any device for heart monitoring purposes.

Using MCG, increased S3 seems to be a plausible marker of the worsening of HF (with the best *p*-value of 0.007). This is in line with the study in [22] regarding implantable devices for HF detection. Instead of diminishing S1, which was observed in [22], we were able to use the temporal changes in a window centered around S1 as a marker of HF; however, the difference between the admission and discharge was statistically less significant. In particular, the stability of S1 was higher in discharge measurements, which means that the correlation between the windows around S1 is heightened. The observed diminishing of S1 and higher stability of S1 during discharge in our study may still be related to each other, since a decrease in S1 amplitude might also contribute to the decreased correlation at admission.

Increased signal RMS strength would seem to be an indicator of the worsening of HF (decompensation). Accelerometer and gyroscope RMS strength both yielded similar and statistically significant *p*-values of 0.033 and 0.03, respectively. This is in line with the results in [16] using slightly different terminology. The advantage of using signal RMS strength against S3 is that it is easier to calculate, as it does not require exact finding of the location around S3. The search for the S3 location might also be more difficult if R-peaks of the ECG are not available, or in the case of AF. The RMS strength was decreased towards discharge in almost as many cases as S3 strength.

Rapid weight gain is considered to be related to the worsening of the HF patient’s condition (at least a few days before hospital admission) [23]. We observed also a weight change from the mean of 85.6 (SD = 26.1) kg at admission to 84.5 (SD = 28.8, NA = 4) kg at discharge (with a *p*-value of 0.005). In total, for 75% (12/16, NA = 4) patients, weight was decreased towards discharge. Whether the patient’s weight or the weight change has any effect on the measured parameters is interesting but might require more data in order to be studied. Potentially adjusting the signal RMS strength (or other features) using the patient’s weight could be useful, as was suggested in [16]. The most probable reason to the weight loss during the hospital period was the usage of diuretics medication to manage the patient’s swelling.

The limitations of this study included the small number of patients (N = 20) and that AF was present in many measurements. For these reasons, the differences between HFpEF and HFrEF could not be identified in terms of signal characteristics. Due to these issues, we decided not to apply supervised ML to the data but rather aimed to find more general concepts which have a physiological meaning. We presented the results in two parts—both with all measurements and with sinus rhythm measurements only—to cope with this, making our results more convincing as well. In the future, a screening tool for predicting HF worsening periods could use a much richer set of features given that there is a sufficient amount of data to examine more sophisticated features. As another limitation of this study, EF was not measured during hospital discharge. Adding this information with more data would allow more advanced analyses.

## 8. Conclusions

We were able to find three features indicating acute decompensated HF in a serial-measurement scenario: an increase in signal RMS strength, an increase in the strength of S3, and a decrease in signal stability around S1. According to our pairwise analysis, these features are all highly discriminative, and potentially, they can be used to complement each other. Limitations of this study included the relatively small number of patients and that some patients had significant arrhythmia. As a summary, we showed that by using a dedicated sensor platform, clinically relevant features which are in line with prior literature can be found to characterize HF using MCG. In the future, MCG could potentially be used non-invasively for the purposes of patient self-screening using either continuous-time or serial measurements with an appropriate device utilizing MCG.

## Figures and Tables

**Figure 1 sensors-22-09781-f001:**
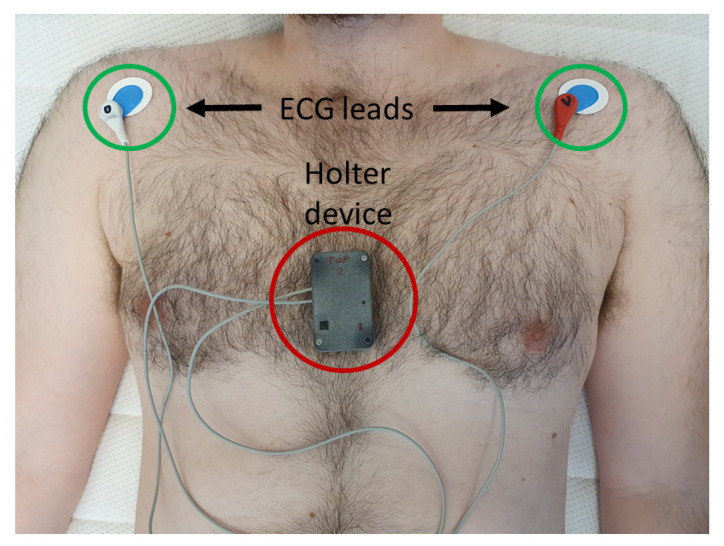
The Holter device is placed on patient’s chest (as shown in the figure) while the patient is in the supine position. It measures 3-axis acceleration and 3-axis rotation, supplemented with one lead ECG.

**Figure 2 sensors-22-09781-f002:**
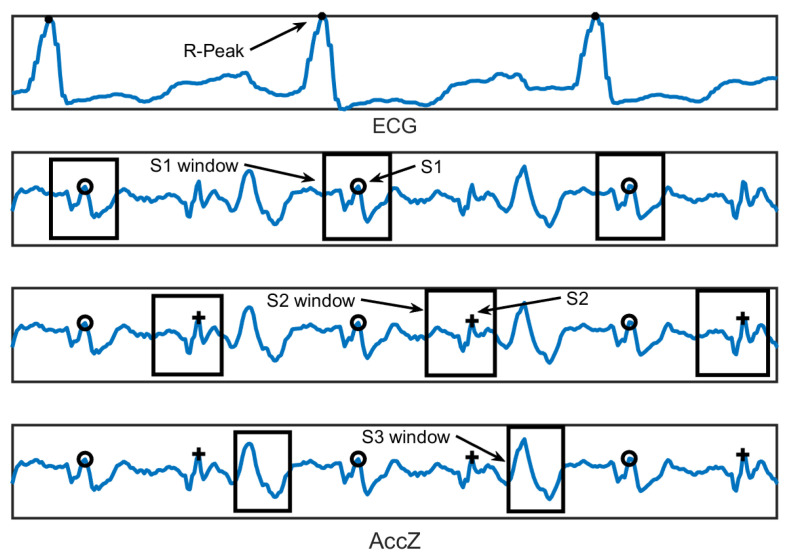
The estimation of locations of S1–S3 from MCG utilizing only the R-peaks of ECG. The plotting indicates the estimated S1 location and window around S1, estimated S2 location and window around S2, and estimated S3 window using AccZ axis.

**Figure 3 sensors-22-09781-f003:**
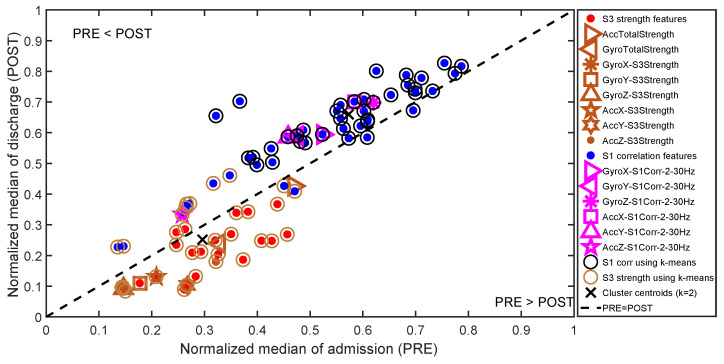
The median value of all signal S3 strength and S1 correlation features during admission (PRE) and during discharge (POST) on the x-axis and y-axis, respectively. Each feature represents a single point marker in the plot (across all admission patients or discharge patients).

**Figure 4 sensors-22-09781-f004:**
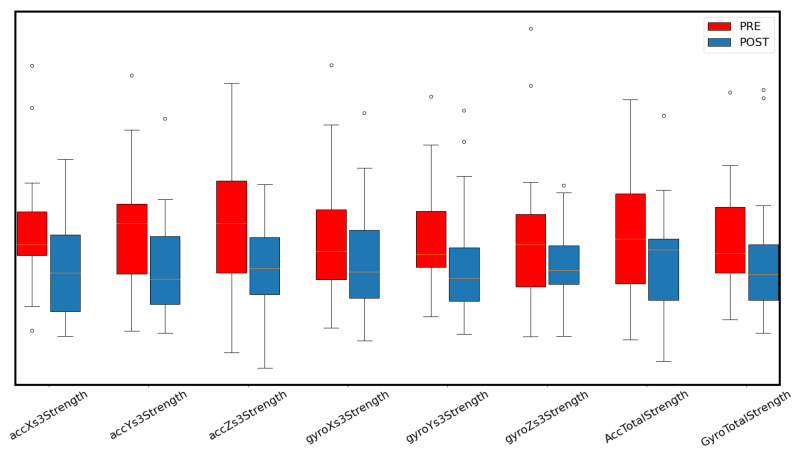
A box plot of selected signal strength features during admission and discharge. The features were normalized pairwise into zero mean and unit standard deviation.

**Table 1 sensors-22-09781-t001:** Comparison of our Holter device to some other reference implementations.

	Sandhi [29]	Yang [25]	Tadi [12]	Ours
Modality	ECG,	ECG,	ECG,	ECG,
	SCG	SCG,	SCG,	SCG,
		GCG	GCG	GCG
Accelerometer	500	256	200	400
Fs Hz				
Accelerometer	120	45	180	25
noise density				
μg/$\sqrt{H}$z				
Gyroscope	N/A	256	200	208
Fs Hz				
Gyroscope	N/A	5	8	7
noise density				
mdps/$\sqrt{H}$z				
Size	3847	62 × 32	50 × 32	59 × 35
mm/mm${}^{2}$	mm${}^{2}$	×12	×11.9	×16
Power mW	7.4	N/A	14.5	28
Weight gm	39	31	N/A	29

**Table 2 sensors-22-09781-t002:** Demographic information of the study.

MECHANO-HF Study	Hospital Admission (N = 20)			Hospital Discharge (N = 20)		
	Mean	STD	N/A	Mean	STD	N/A
Heart rate (ECG, bpm)	86.0	19.1	0	82.4	17.6	0
Weight (kg)	85.6	26.2	0	84.5	28.8	4
Diastolic BP (mmHg)	70.1	18.5	0	67.0	15.8	12
Systolic BP (mmHg)	129.4	25.5	0	117.9	17.4	12
Ejection fraction (%)	33.9	12.3	4	N/A	N/A	20
ProBNP (ng/l)	7755	4494.4	14	N/A	N/A	20
NYHA class (I-IV,1-4)	3.2	0.67	6	2.7	0.76	13

**Table 3 sensors-22-09781-t003:** Signal strength around third heart sound (S3) and accelerometer and gyroscope overall RMS strength. Signal strength tended to decrease from admission to discharge. The estimation of S3 is more inaccurate in AF measurements, and therefore, results with sinus rhythm only were also added to the bottom row. The Wilcoxon *p*-values are paired and two-sided.

	AccX S3 Strength	AccY S3 Strength	AccZ S3 Strength	GyroX S3 Strength	GyroY S3 Strength	GyroZ S3 Strength	Acc RMS Total Strength	Gyro RMS Total Strength
Mean (Admission)	1049.7	964.3	1388.4	206.5	173.0	67.0	1900.7	201.3
Mean (Discharge)	863.6	766.9	1109.9	165.8	140.4	54.5	1683.3	175.6
Median (Admission)	974.6	1002.2	1432.6	181.5	149.4	62.2	1881.7	182.5
Median (Discharge)	829.8	693.4	1098.1	150.5	122.3	51.5	1775.6	156.7
STD (Admission)	305.6	348.6	507.5	96.2	64.8	29.6	595.1	63.2
STD (Discharge)	283.3	287.6	322.9	81.7	70.9	17.2	515.1	77.8
Wilcoxon *p*-value	0.025	0.007	0.012	0.015	0.007	0.117	0.033	0.030
Decrease to	N = 14/20,	N = 16/20,	N = 16/20,	N = 17/20,	N = 17/20,	N = 13/20,	N = 16/20,	N = 16/20,
discharge (N)	all	all	all	all	all	all	all	all
	N = 9/11,	N = 9/11	N = 9/11	N = 10/11	N = 10/11	N = 6/11	N = 9/11,	N = 10/11,
	SR only	SR only	SR only	SR only	SR only	SR only	SR only	SR only

**Table 4 sensors-22-09781-t004:** Mean of signal stability (corr.) around S1 (first heart sound) in the 2–30 Hz frequency band. It can be observed that the S1 correlation is increased towards discharge. This frequency band was selected to provide consistent results, although other similar windows worked as well. The estimation of correlation is more inaccurate in AF measurements, and therefore, results with sinus rhythm only were also added to the bottom row. The Wilcoxon p-values are paired and two-sided.

	AccX corr. S1 2–30 Hz	AccY corr. S1 2–30 Hz	AccZ corr. S1 2–30 Hz	GyroX corr. S1 2–30 Hz	GyroY corr. S1 2–30 Hz	GyroZ corr. S1 2–30 Hz
Mean (Admission)	0.705	0.670	0.366	0.560	0.634	0.746
Mean (Discharge)	0.754	0.717	0.352	0.601	0.677	0.788
Median (Admission)	0.729	0.648	0.283	0.541	0.604	0.776
Median (Discharge)	0.798	0.725	0.343	0.592	0.679	0.817
STD (Admission)	0.177	0.190	0.231	0.199	0.192	0.144
STD (Discharge)	0.147	0.143	0.189	0.176	0.191	0.149
Wilcoxon *p*-value	0.279	0.179	0.433	0.179	0.232	0.156
Increase to	13/20,	15/20,	14/20	13/20	14/20,	15/20,
discharge (N)	all	all	all	all	all	all
	7/11,	9/11,	8/11,	7/11	8/11	9/11
	SR only	SR only	SR only	SR only	SR only	SR only

## Data Availability

The data used in the study is governed by Finnish law of confidential clinical data and will not therefore be made available publicly.

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
