# Peer review of "Mechanocardiography-Based Measurement System Indicating Changes in Heart Failure Patients during Hospital Admission and Discharge"

_sensors, 2022, doi:10.3390/s22249781_

Round 1

Reviewer 1 Report

- From Table 2 we notice quite high STD of weight; I suppose the distribution in the subjects may be wide as well. Please comment on the implications of different weight and age on the HF diagnostic parameters which you measure,
- What does “NA” mean in the Table 2?
- In line 178 please remove extra “to”,
- Please comment why the recordings from different subjects originally were of different duration. Was any length of corrupted recording set as a threshold for exclusion of the entire recording?
-  Please mention if you plan to make the code and dataset available for public

Author Response

  1. From Table 2 we notice quite high STD of weight; I suppose the distribution in the subjects may be wide as well. Please comment on the implications of different weight and age on the HF diagnostic parameters which you measure.

We estimate that the interpersonal changes for the weight don't affect the measurements unless the weight changes for a person between the measurements. This is because we use serial measurements in this study.

  1. What does “NA” mean in the Table 2?

NAs in table 2 have been corrected to N/A which means the number of not known values in each row.

  1. In line 178 please remove extra “to”

In response to the reviewer’s suggestion, we reviewed the whole paper to fix all the typos and grammatical errors including this specific one mentioned by the reviewer. 

  1. Please comment why the recordings from different subjects originally were of different duration. Was any length of corrupted recording set as a threshold for exclusion of the entire recording?

The first 12 recordings were recorded with a length of approximately 8-9 minutes, but for the following 8 recordings, the length was decreased to approximately half of that. The first 12 recordings were done during the first phase of the trial and the 8 remaining recordings were added later to include more patients. We did not include any initial limit to exclusion of the entire recording, but the length/quality of the recordings after MA removal was seen as sufficient for inclusion in the study.

  1. Please mention if you plan to make the code and dataset available for public.

We are not planning to make the data public.

Reviewer 2 Report

The authors aimed to assess which kind of person-specific changes can be observed between the signals acquired during acute decompensation state and normal state in heart failure (HF) patients. Thus, the current work tried to build a ground for effective screening of HF reducing hospitalizations and the treatment cost in the future. The study sparks interest and has a potential contribution to this field. However, I have minor comments to address.

How does the present device differ from an ECG with an Accelerometer, already commercially available?

How could this device innovate from a clinical point of view?

Can this equipment also be used for other populations?

Author Response

  1. How does the present device differ from an ECG with an Accelerometer, already commercially available?

The main difference is that the device has been used in this study has also the Gyro data in use as mentioned. In addition, our method is meant to use in spot measurements and that's why it's beneficial for many use cases.

  1. How could this device innovate from a clinical point of view?

This device can reduce healthcare costs and paves the way for follow-up measurements as well as patient telemonitoring as mentioned in the introduction section. For example, it can help prevent the decompensation state.  

  1. Can this equipment also be used for other populations?

This device can be used probably for other populations, but it requires future research. At the moment, the targeted population is HF patients.

- Also regarding "Moderate English changes required":

In response to the reviewer’s suggestion, we reviewed the whole paper to fix all the typos and grammatical errors.